# Skipping the Frame-Level: Event-Based Piano Transcription With Neural Semi-CRFs

**Yujia Yan**∗, **Frank Cwitkowitz, Zhiyao Duan**
Department of Electrical and Computer Engineering
University of Rochester
Rochester, NY, 14627 USA
`yujia.yan@rochester.edu, fcwitkow@ur.rochester.edu`
`zhiyao.duan@rochester.edu`

## Abstract

Piano transcription systems are typically optimized to estimate pitch activity at each frame of audio. They are often followed by carefully designed heuristics and post-processing algorithms to estimate note events from the frame-level predictions. Recent methods have also framed piano transcription as a multi-task learning problem, where the activation of different stages of a note event are estimated independently. These practices are not well aligned with the desired outcome of the task, which is the specification of note intervals as holistic events, rather than the aggregation of disjoint observations. In this work, we propose a novel formulation of piano transcription, which is optimized to directly predict note events. Our method is based on Semi-Markov Conditional Random Fields (semi-CRF), which produce scores for intervals rather than individual frames. When formulating piano transcription in this way, we eliminate the need to rely on disjoint frame-level estimates for different stages of a note event. We conduct experiments on the MAESTRO dataset and demonstrate that the proposed model surpasses the current state-of-the-art for piano transcription. Our results suggest that the semi-CRF output layer, while still quadratic in complexity, is a simple, fast and well-performing solution for event-based prediction, and may lead to similar success in other areas which currently rely on frame-level estimates.

## 1 Introduction

The task of Automatic Music Transcription (AMT) aims to transcribe a music recording into some form of music notation [Benetos et al., 2018]. Examples of notation include MIDI event sequences, e.g, Hawthorne et al. [2018], Kong et al. [2020], Kim and Bello [2019], Kwon et al. [2020], and staff notation, e.g., Nakamura et al. [2018], Román et al., 2019]. In this work, we address the problem of transcribing piano music into a MIDI event sequence. MIDI transcription involves constructing a sequence of events, each specified by its onset and offset positions, with the constraint that two events of the same event type (e.g., certain pitches and pedals) cannot overlap. In addition to onsets and offsets, the velocity (i.e., a value that represents the intensity of a key strike, which informs the loudness) associated with each event is often estimated.

In recent years, neural network based approaches have reached the state of the art for the problem of piano transcription, e.g., Hawthorne et al. [2018], Kong et al. [2020], Kwon et al. [2020]. They operate at the frame-level and make predictions for different stages of a note event, i.e., the onset, offset, and pitch activation, separately. In order to extract note-level predictions, they use manually designed procedures to combine the disjoint frame-level predictions. These include thresholding,

---

∗Code is available at `https://github.com/Yujia-Yan/Skipping-The-Frame-Level`

35th Conference on Neural Information Processing Systems (NeurIPS 2021).

peak picking [Hawthorne et al., 2018, Kong et al., 2020] or low-order hidden Markov models [Kwon et al., 2020]. This two-stage approach requires manually crafted procedures and a manually designed state structure modeling the temporal evolution of notes.

In this work, we propose a direct approach to note-level transcription. Instead of scoring and aggregating note activations across individual time frames, our approach directly scores a time interval's likelihood of covering the entire process of a certain event (e.g., a note, pedal usage). This greatly simplifies the formulation of piano transcription. Specifically, the proposed method takes the log-mel spectrogram of an audio segment (e.g., 10s) as input, and uses a contextual model to produce contextual embeddings for each frame. It then uses a score model to score each possible time interval within the segment to assess whether it covers the entire span of a musical event (e.g., notes, pedals) as specified by a specialized zeroth-order semi-Markov conditional random fields (semi-CRF) [Sarawagi and Cohen, 2004]. These scores are then decoded using Viterbi algorithm into a set of non-overlapping intervals for each event type, i.e., note-level transcription. Finally, other attributes of these intervals (i.e., velocity and refined boundary positions) are also estimated.

Previous works using semi-CRF were concerned with shorter sequence lengths and smaller interval lengths [Kong et al., 2016, Liu et al., 2016, Kemos et al., 2019, Lu et al., 2016], and usually only attempted to estimate a single channel (track) of non-overlapping events. In contrast, our problem deals with sequences of 400-1600 frames (approximately 10s-40s) and requires a separate semi-CRF for around 90 different labelling channels (88 pitches + 1-3 pedals), each corresponding to a specific event type. Another challenge is the wide range of event duration that can be encountered in piano music, which further increases the computational expenses of the problem. These challenges prevented us from using existing general purpose semi-CRF methods due to computational expenses. Instead, we introduce a zeroth-order Semi-CRF formulation that is specifically adapted to the problem and show that it can be implemented efficiently (see Table 3) for the problem size of interest.

Our experiments on the MAESTRO dataset show that the proposed system achieves *Note w/ Offset* $F_1$ of 88.72% and *Note w/ Offset & Velocity* $F_1$ of 87.75%, exceeding the current state of the art while being smaller and faster. We believe that this simple, fast and well-performing approach is also extensible to other similar tasks with intervals as the prediction target, such as polyphonic sound event detection or speaker diarization.

## 2 Related Works

**Piano Transcription** There have been many proposed approaches to piano transcription in the last several decades. Early works consist of simple signal processing methods such as spectral peak-picking [Klapuri et al., 2000, Bello et al., 2006], which estimates fundamental frequencies directly from the spectrum, or spectral decomposition [Smaragdis and Brown, 2003, O'Hanlon and Plumbley, 2014], where a short-time spectrum is factorized into spectral components with corresponding activations. Several parametric models have also been proposed to perform spectro-temporal decomposition [Vincent et al., 2009, Emiya et al., 2009, Cheng et al., 2016]. Time-domain decomposition has been also investigated in Cogliati et al. [2017, 2016]. Most of these methods aim to model notes explicitly as parametric templates, and tend to suffer from the lack of generalization. Other methods include machine learning techniques such as Hidden Markov Models [Raphael, 2002, Böck and Schedl, 2012] or Support Vector Machines [Poliner and Ellis, 2006, Weninger et al., 2013]. Recently, neural network based approaches have made significant advances [Sigtia et al., 2016, Hawthorne et al., 2018, 2019, Kim and Bello, 2019] by taking advantage of large Disklavier piano databases.

Many methods separate piano transcription into frame-wise polyphonic pitch estimation and note tracking, thereby optimizing for frame-level predictions which are used to estimate quantized note intervals. While intuitive, this practice has a major drawback: the quality of the note-level predictions is fundamentally limited by the quality of the frame-level activations, which can be sporadic, discontinuous, and non-uniform in terms of their strength. Kameoka et al. [2007] attempted to circumvent this problem by performing harmonic temporal template structured clustering to explain spectrogram observations as originating from distinct sources, directly estimating notes as discrete events. This reflects the idea of [Bregman, 1990], i.e., that meaningful auditory events are results of simultaneous and sequential groupings. However, the underlying mental process of low-level groupings are more complex than clustering according to simple parametric harmonic temporal templates.

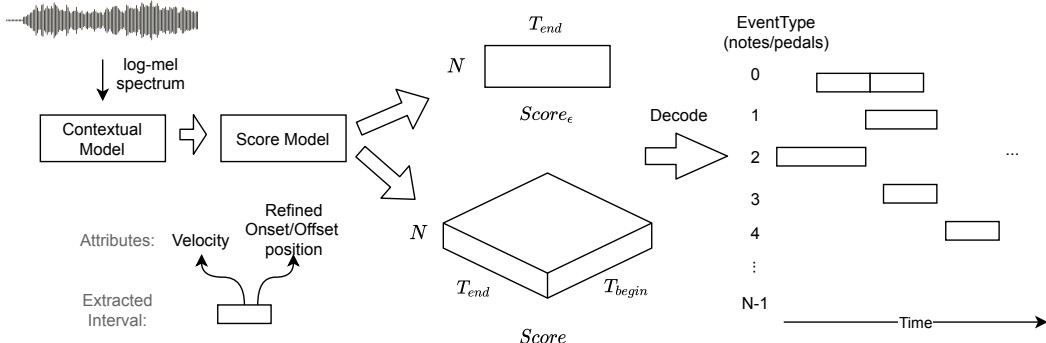

Figure 1: Proposed system overview. In the middle part of the figure, $T_{begin}$ and $T_{end}$ are the number of beginning positions and ending positions, respectively, and $N$ is the number of *eventType*(s).

Some more recent works [Kelz et al., 2019, Kwon et al., 2020] attempt to incorporate manually-defined state structure for modeling the temporal evolution of notes. However, the state structure (usually of low order) may not reflect the actual temporal evolution and thus limits the representative power of these models. Cogliati et al. [2017] directly estimates the activation of note templates in the time domain, but these templates are preset and are piano- and context-dependent, lacking the generalization to unseen pianos or recording environments.

The proposed method learns to estimate note events holistically, rather than adopting the frame-to-note two-stage approach or performing and aggregating multiple subtasks (i.e., onset, offset, and pitch estimation). It can be viewed as performing groupings directly in the event space, with low-level details being learned by the neural networks.

**Semi-Markov CRFs**   A semi-Markov conditional random field (semi-CRF, Sarawagi and Cohen [2004]) defines a conditional probability distribution over sets of non-overlapping labeled intervals within an input sequence. Semi-CRFs have been used in Chinese word segmentation [Liu et al., 2016, Kong et al., 2016], named entity recognition [Zhuo et al., 2016, Ye and Ling, 2018, Arora et al., 2019], character-level part-of-speech tagging [Kemos et al., 2019], phone recognition [Lu et al., 2016], chord recognition [Masada and Bunescu, 2017], etc. The time complexity for computing the partition function and inferring the most likely configuration is quadratic with respect to the length of the sequence. Therefore, most works set an upper bound on the length of a single interval, e.g., Kemos et al. [2019], to make the computation tractable.

In this work, a semi-CRF is defined for each event type (i.e., a note of a specific pitch, a pedal). Our semi-CRF formulation differs from the standard formulation in the following ways: 1) Different events in a semi-CRF are allowed to overlap on their endpoints (boundary frames), since audio frames represent a time period and can contain both the end and beginning of an event of the same type. This treatment allows us to use a larger hop size without adjustment due to overlap in the same frame, so that the sequence length for the same audio segment can be reduced; 2) A frame is not required to belong to any event. Despite the loosened definition, we use the name semi-CRF to refer to this type of structured prediction module.

Compared to the tasks mentioned above, the task of piano transcription has a longer input sequence and a larger range of possible event duration. The former makes the time complexity issue more prominent, while the latter makes it difficult to specify an upper bound on event duration for speedup purposes. We show that our formulation is efficient for piano transcription and hence does not require an upper bound on event duration, thanks to a set of implementation optimizations.

## 3   Proposed Semi-CRF Approach to Piano Transcription

The proposed system transcribes the input audio into a list of musical events, i.e., notes and pedals. Here the term *events* refers to acoustical events that span certain time intervals (not to be confused with raw MIDI events, i.e., *note on*, *note off*, *cc*, etc., used for serializing the performance). Taking

an audio segment (e.g., 10s), the transcription process is illustrated in Figure 1. First, a log-mel spectrogram is computed after taking the short-time Fourier transform (STFT) of the audio. This is used as input to a *contextual model* to produce contextual features across time frames. Such features are then fed to a *score model* to calculate two kinds of scores. The first kind of score ($Score$) assigns a score to indicate whether an interval is an event of a certain event type. The second kind of score ($Score_\epsilon$) indicates whether an interval that spans two frames is not part of any event for a certain event type. Finally, a Viterbi algorithm is used to decode the most likely sequence of events for each event type, using the aforementioned scores. Attributes, i.e., velocity and refined onset/offset positions, are then estimated for each extracted event.

### 3.1 CRF Formulation

Let $\mathcal{X} = <x_0, x_1, \ldots, x_{N-1}>$ be an audio segment containing a sequence of $N$ time frames. Let $\mathcal{Y} = \{(i, j, eventType), i \leq j\}$ be the set of musical events entirely contained within this segment, with time quantized to audio frames. Here $i$ and $j$ are respectively the beginning and ending frame indices for each event, and *eventType* is the type of the event, e.g., a specific key (pitch) of the the 88 keys of a piano or the sustain pedal. Events that extend outside the audio segment are not considered in this formulation, but will be handled in the inference process (See Section 3.3). We assume that for the same event type, two events $A$ and $B$ are non-overlapping, i.e., either $j_A \leq i_B$ or $j_B \leq i_A$. In this formulation, it is allowed to have single-frame events where $i = j$ (note how this would make the formulation differ from the standard semi-CRFs).

We associate each event with a set of attributes: 1) the relative non-quantized position of onsets and offsets at the sub-frame level, represented by a value from $-0.5$ to $0.5$ indicating the position relative to the quantized index of the frame, and 2) the MIDI velocity of the event, represented by a discrete value from 0 to 127.

We use $\mathcal{Y}_{eventType}$ to denote the subset of events that contains only a specific event type. For each event type, we model the following conditional probability:

$$p(\mathcal{Y}_{eventType}|\mathcal{X}) = \frac{1}{Z(eventType)} \exp \left[ \sum_{(i,j,eventType)\in\mathcal{Y}_{eventType}} score(i, j, eventType) \right. \\ \left. + \sum_{[i-1,i] \text{ not covered in } \mathcal{Y}_{eventType}} score_\epsilon(i - 1, i, eventType) \right], \tag{1}$$

where $score(i, j, eventType)$ assigns a score to the interval $[i, j]$ to indicate whether it is an event of *eventType*, $score_\epsilon(i - 1, i, eventType)$ assigns a score to the interval $[i - 1, i]$ that spans two frames to indicate whether it is not covered by any event of *eventType*, which inversely represents the frame-level activation for an event, and $Z(eventType)$ is the normalization factor. The non-event score ($score_\epsilon$) for two consecutive frames comes from skipping one position in a feasible solution (see example in the next paragraph), and it also allows the model to degenerate to frame-level event prediction. Here, for notational convenience, we omit $\mathcal{X}$ for every term.

For numerical stability, Eqn. (1) is computed in the log-domain. The exponent, i.e., the summation of all $score(i, j, eventType)$ and $score_\epsilon(i - 1, i, eventType)$ corresponding to $\mathcal{Y}_{eventType}$, is the unnormalized log-likelihood. As an example, for a single *eventType*, assuming there are 7 audio frames <0,1,2,3,4,5,6> in total, for the interval set candidate {[0,0], [2,4], [4,5]}, the corresponding unnormalized log-likelihood is computed as $score(0, 0) + score_\epsilon(0, 1) + score_\epsilon(1, 2) + score(2, 4) + score(4, 5) + score_\epsilon(5, 6)$.

The computation of $\log Z(eventType)$ and its gradient w.r.t. to $score/score_\epsilon$, $\nabla \log Z(eventType)$, is critical in training. The detailed procedure is shown in Algorithm 1 (forward-backward algorithm). In practice, we compute $\log Z$ and $\nabla \log Z$ for all *eventType*(s) in parallel. The forward stage and the backward stage in the forward-backward algorithm are batched together, but with all positions of the input flipped, as to compute them in a single pass without the need for two separate update equations, since their calculations are essentially the same. We also find that a substantial speedup (>5x) can be achieved by using a custom gradient computation via the backward pass of the forward-backward algorithm for $\log Z$, as opposed to using automatic differentiation w.r.t. $\log Z$ in *PyTorch*. This is likely due to a more contiguous memory access pattern and the elimination of unnecessary operations.

In order to make memory access at each step more contiguous, the event scores $Score$ and $Score_\epsilon$ are organized as $T_{end} \times T_{start} \times N$ and $T_{end} \times N$ tensors, respectively, with $T_{end}$ being the ending position of an interval, $T_{start}$ being the beginning position of an interval, and $N$ being the number of *eventType*(s). Here, each slice along the $N$ dimension represents a separate semi-CRF channel.

A runtime benchmark of the semi-CRF layer presented is provided in Table 3.

---

**Algorithm 1** Forward-backward algorithm for $\log Z$ and $\nabla \log Z$ for a specific event type.

---

**Input:** function $score(i,j)$, function $score_\epsilon(i-1,i)$
**Output:** $\log Z$ and $\nabla \log Z$
  **Forward stage:**
  Initialize the forward variable: $v(0) \leftarrow \log(\exp(score(0,0)) + 1)$
  **for all** $j = 1, \ldots, N-1$ **do**

$$v(j) \leftarrow \log \left\{ \exp[v(j-1) + score_\epsilon(j-1,j)] + \sum_{k<j} \exp[v(k) + score(k,j)] \right\}$$

$$v(j) \leftarrow v(j) + \log\{1 + \exp[score(j,j)]\}$$

  **end for**

  **Readout the log partition function:**
  $\log Z \leftarrow v(N-1)$

  **Backward stage:**
  Initialize the backward variable: $q(N-1) \leftarrow \log\{\exp[score(N-1,N-1)] + 1\}$
  **for all** $j = N-2, \ldots, 0$ **do**

$$q(j) \leftarrow \log \left\{ \exp[q(j+1) + score_\epsilon(j,j+1)] + \sum_{k>j} \exp[q(k) + score(j,k)] \right\}$$

$$q(j) \leftarrow q(j) + \log\{1 + \exp[score(j,j)]\}$$

  **end for**

  **Read out the posterior marginals as derivatives:**
  **for all** i=0,..., N-1 **do**
    $p(i,i) \leftarrow \exp\{v(i) + q(i) + score(i,i) - 2\log[\exp(score(i,i) + 1)] - \log Z\}$
  **end for**
  **for all** i<j **do**
    $p(i,j) \leftarrow \exp[v(i) + q(j) + score(i,j) - \log Z]$
  **end for**
  **for all** i=1,..., N-1 **do**
    $p_\epsilon(i-1,i) \leftarrow \exp[v(i-1) + q(i) + score_\epsilon(i-1,i) - \log Z]$
  **end for**
  $\frac{\partial \log Z}{\partial score(i,j)} = p(i,j), \frac{\partial \log Z}{\partial score_\epsilon(i-1,i)} = p_\epsilon(i-1,i)$

---

### 3.2 Training Objectives

For training, we use maximum likelihood estimation (MLE), where the conditional log-likelihood is defined to consolidate the conditional probability in Eqn. (1) over all event types, assuming their conditional independence given $\mathcal{X}$:

$$\log p(\mathcal{Y}|\mathcal{X}) = \sum_{eventType} \log p(\mathcal{Y}_{eventType}|\mathcal{X}). \tag{2}$$

One may question the validity of this conditional independence assumption on different *event-Type*(s); here, we keep things simple, as previous approaches often make this assumption. We leave this for future investigation.

In addition to the log-likelihood for the presence of events defined in Eqn. (2), we also learn to predict three attributes for each event:

$$\log p(\textit{attributes}|e) = \log p(\textit{velocity}|e) + \log p(\textit{refined onset}|e) + \log p(\textit{refined offset}|e). \quad (3)$$

Here we use $e$ to denote an event. We parameterize these terms with the following distributions, i.e., the Softmax/Multinomial distribution, and the Continuous Bernoulli distribution [Loaiza-Ganem and Cunningham, 2019]:

$$\textit{velocity}|e \sim \text{Softmax}(\mu(e)),$$
$$0.5 + \textit{refined onset/offset}|e \sim \text{ContinuousBernoulli}(\lambda(e)), \quad (4)$$

where $\mu(e)$ and $\lambda(e)$ are parameters produced by neural networks that take features of the interval as the input. The interval features used in this work are described in Section 3.4. The final objective is defined as

$$\mathfrak{L} = -[\log p(\mathcal{Y}|\mathcal{X}) + \sum_{e \in \mathcal{Y}} \log p(\textit{attributes}|e)]. \quad (5)$$

More discriminative and cost-sensitive losses such as max-margin and softmax margin can be used as drop-in replacements, but we leave this investigation to future work.

### 3.3 Inference

We use dynamic programming (Viterbi) to infer the most likely interval sequence for every *eventType* independently. The procedure is described in Algorithm 2.

When processing longer audio recordings, our system transcribes audio segment by segment. Because the system is designed to ignore all events that extend outside a segment, these audio segments need to overlap with each other. In this work, segments are 10s long and the overlap is 5s. In order to properly handle notes near the boundaries of a segment, we modify the algorithm such that it is forced to take the result from the overlapping portions into account as follows. Within an audio segment, decoding is performed in reverse order: backtracking starts from the position immediately after the last event decoded in the last segment of the same event type, if any of those overlap with the current segment. We found that, for each segment, it works slightly better to discard all events of which the onsets fall into the overlapping region with the next segment, which is a procedure similar to the *overlap-discard* method for computing discrete convolutions for a long signal.

---

**Algorithm 2** Viterbi (MAP) decoding of a specific *eventType* within an audio segment.

---

**Input:** function $score(i, j)$, function $score_\epsilon(i - 1, i)$, backtracking starting frame position $t$
**Output:** a set of intervals $\mathcal{Y}$
$\quad v(N - 1) \leftarrow \max(score(N - 1, N - 1), 0)$
$\quad$**for all** $j \in N - 2, \ldots, 0$ **do**
$$\quad\quad v(j) \leftarrow \max \begin{cases} v(j + 1) + score_\epsilon(j, j + 1) \text{ - \textit{skip if inactive}} \\ \max_{k > j}\{v(k) + score(j, k)\} \text{ - \textit{if an interval}} \end{cases}$$
$\quad\quad v(j) \leftarrow v(j) + \max(score(j, j), 0)$ - *single frame case*
$\quad$**end for**
$\quad$Perform backtracking starting from position $t$ to get $\mathcal{Y}$

---

After events are extracted, feature vectors of the events, described in Section 3.4, are used to predict attributes, namely, velocity and refined onset/offset positions. These attributes are then assembled with the events to form an event tuple (*onset*, *offset*, *eventType*, *velocity*) for the final output.

### 3.4 Model Architectures

In this section, we present details of the neural architectures for the three components used in the proposed approach: 1) contextual model, 2) score models, and 3) attribute predictors. To recap, we first apply a contextual model to transform the input audio frames into a sequence of contextual embeddings. These contextual embeddings are then used by the score models to compute $score(\cdot)$ for all possible events, in a fashion similar to Cross and Huang [2016], Kitaev and Klein [2018], Liu et al. [2016], and $score_\epsilon(\cdot)$. After events are extracted, contextual embeddings are also used by the attribute predictors to predict attributes (i.e., velocity and refined onset/offset times) associated with each event.

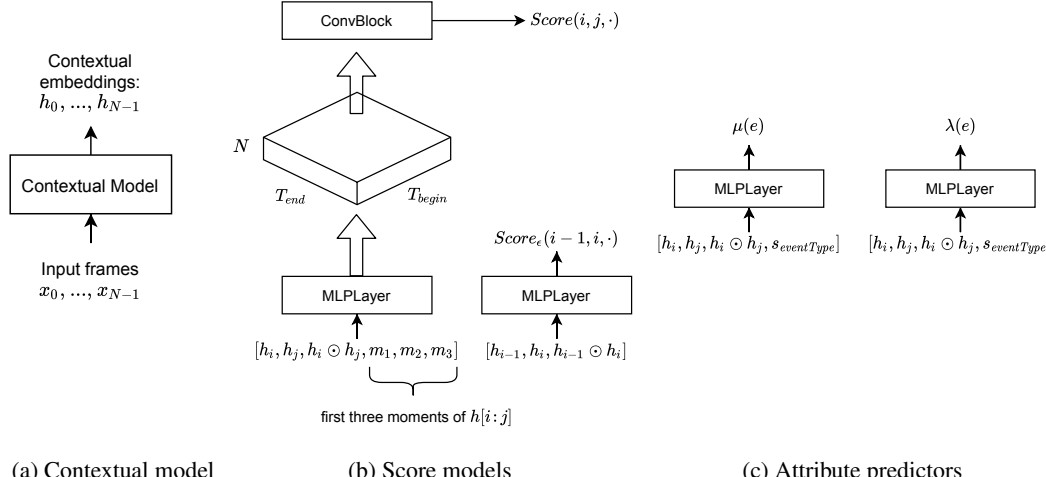

(a) Contextual model        (b) Score models        (c) Attribute predictors

Figure 2: Model architectures. The contextual model transforms the input frame sequence into a sequence of context embeddings, based on which scores and attributes are computed by the score models and attribute predictors, respectively.

### 3.4.1 Contextual Model

As input to the contextual model, we compute a log-mel spectrogram with a frame size of 4096 samples, a hop size of 1024 samples, and the Hann window function. Following Hawthorne et al. [2018], Kong et al. [2020], we use 229-bands with a frequency range from 30 Hz to 8000 Hz. In addition to a single Hann window applied to each frame, we also experimented with applying multiple adjustable Gaussian windows (5 windows, initialized with equidistant centers and constant variance) to the same frame and concatenating the resulting log-mel spectra together for improving the temporal resolution of the spectra input. By doing this, the input becomes a log-mel spectrogram with multiple channels for each time-frequency bin, with each channel obtained from a different window function. We feed the log-mel spectrogram through four convolutional blocks with a kernel size of 3 and 48, 64, 92, 128 filters, respectively. Each convolutional block contains two 2-d convolution layers, followed by batch normalization and a Gaussian Error Linear Unit (GELU) activation function [Hendrycks and Gimpel, 2016]. At the end of each block, the output is pooled along the frequency dimension using average pooling with a kernel size and stride of 2. The channel and frequency dimensions are flattened together and fed into a two-layer bidirectional GRU with hidden size 256. This contextual model is largely the same as the one used in Kong et al. [2020], however, we only utilize one instance of the block rather than stacking it for different frame-level prediction targets.

### 3.4.2 $Score(i, j, \cdot)$ and $Score_\epsilon(i-1, i, \cdot)$

As shown in Fig. 2b, the features used for scoring an interval $[i, j]$ include contextual embeddings at its two endpoints, i.e., $\mathbf{h}_i$, $\mathbf{h}_j$, their elementwise multiplication $\mathbf{h}_i \odot \mathbf{h}_j$, and the first three moments for the contextual embeddings within the interval $[i, j]$. These features are chosen because they provide information for events as a whole while being cheap to compute, and are often seen in the literature when extracting holistic/pairwise features. These features are concatenated and fed into a three-layer feed-forward neural network (*MLPLayer*) to obtain a raw intervalic score tensor with shape $T_{end} \times T_{begin} \times N$, where the three dimensions are the ending position of an interval, the beginning position of an interval, and *eventType* channels, respectively. The output size of this network is equal to the number of event types and the hidden size is equal to 4x the output size. On top of the raw score tensor, as shown in the figure, we experimented with applying a simple convolutional block with two convolutional layers of kernel size 3 and filter size being 3x the output size, with the hope that the block can aggregate neighboring information in the space of intervals in order to increase the expressivity. We also experimented with scaling the intervallic score by the length of the interval, motivated by the following observation: when the score function for an interval is chosen to be the variance alone with this length scaling added, the semiCRF layer in this

work is equivalent to optimal 1-d k-means clustering. This scaling adjusts the score for an event to be roughly at the same scale of the aggregated non-event score at the same time interval at initialization.

For the inactivity score, $score_\epsilon(i-1, i)$, only contextual embeddings of two consecutive frames are used. These are similarly fed into a three-layer feed-forward network with an output size equal to the number of event types.

### 3.4.3 Attribute Predictor

As shown in Fig. 2c, for predicting attributes associated with each event, we use features of end-points concatenated with an trainable *eventType* embedding vector $s_{eventType}$ with size 256 for fully specifying the event. These features are fed into separate multi-layer feed-forward networks. The velocity prediction network has hidden sizes of 512/512, and produces a 128-dimensional logits to infer the MIDI velocity value. The refined onset and offset time prediction network has hidden sizes of 512/128, and produces a 2-dimensional real vector, which is used as logits for the Continuous Bernoulli Distribution for onset/offset refinement, respectively, as defined in Eqn. (4).

## 4 Experiments

### 4.1 Dataset

We conduct our experiments using the MAESTRO v2 dataset [Hawthorne et al., 2019], which contains around 200 hours of MIDI-synchronized (3ms precision) virtuoso piano performance recordings. The recordings were collected across several years of the *International Piano-e-Competition*, and were recorded on *Yamaha Disklavier* pianos. All recordings are sampled at 44.1 kHz, except for files from the 2017 and 2018 competitions, which are sampled at 48 kHz; we thus downsample them to 44.1 kHz for consistency. For comparing with other works, we follow the convention to extend the offset of notes to the offset of any simultaneous sustain pedal event [Hawthorne et al., 2018, Kong et al., 2020].

### 4.2 Training

We use a batch size of 12 and Adabelief [Zhuang et al., 2020] optimizer with a weight decay of 1e-4. We use oneCycle [Smith and Topin, 2019] learning rate scheduler with maximum learning rate set to 6e-4 for 180k iterations and cosine annealing. The learning rate is increased gradually for 20% of iterations and then gradually annealed to 1.5e-5. We automatically determine the value for gradient clipping by using the 0.8 quantile of the gradient norm during the last 10k iterations, which is a strategy similar to Seetharaman et al. [2020]. We apply dropout with rate 0.1 on the attribute predictors and the score model.

### 4.3 Evaluation Metrics

We follow the standard piano transcription evaluation procedure, validating note predictions with multiple levels of criteria. The most basic metric considers a note prediction correct if the estimated onset is within 50 ms of the corresponding ground-truth onset. The next incremental metric additionally requires that the offset prediction is within 50 ms or 20% of the note duration from the ground-truth offset position. The final incremental metric additionally requires that the velocity estimate is within a tolerance of 0.1 ([0,1] normalized velocity) from the ground-truth velocity, as defined in Hawthorne et al. [2018]. We directly use the implementation contained in the *mir_eval* library [Raffel et al., 2014] to compute these three note-level metrics.

We also introduce activation-level metrics which serve as a hopsize-agnostic replacement for the commonly reported frame-level metrics for evaluating how predicted time spans of events overlap with the ground truth (see the supplementary material for details).

Following the convention, each result is averaged across all pieces within the test set. We compute similar metrics, minus the velocity variation, for the sustain pedal activity, as in Kong et al. [2020].

## 4.4  Main Results

We compare the proposed system to the state-of-the-art methods[2] , for piano transcription using the MAESTRO v2 test split. We recompute these metrics for other systems directly from the transcribed MIDI files generated by their pretrained models. We also report our results for our model trained and evaluated on the MAESTRO v3 splits for future reference. The note transcription and pedal transcription results are listed in Tables 1 and 2, respectively.

| Method | Activation | | | Note Onset | | | Note w/ Offset | | | Note w/ Offset & Vel. | | |
|---|---|---|---|---|---|---|---|---|---|---|---|---|
| | P | R | $F_1$ | P | R | $F_1$ | P | R | $F_1$ | P | R | $F_1$ |
| MAESTRO v2 | | | | | | | | | | | | |
| Hawthorne et al. [2019] | 86.84 | 89.24 | 87.82 | 97.88 | 92.26 | 94.93 | 82.09 | 77.44 | 79.65 | 78.37 | 73.94 | 76.05 |
| Kong et al. [2020] | 90.09 | **90.42** | 90.15 | 98.16 | **95.46** | **96.77** | 85.65 | 83.32 | 84.45 | 84.18 | 81.92 | 83.02 |
| Proposed w/o Extra Win | 93.84 | 88.07 | 90.75 | 98.85 | 93.97 | 96.31 | 90.67 | 86.24 | 88.37 | 89.60 | 85.25 | 87.34 |
| Proposed w/o Post Conv | **93.90** | 88.18 | 90.85 | **98.86** | 94.06 | 96.36 | 90.52 | 86.18 | 88.26 | 89.49 | 85.23 | 87.27 |
| Proposed w/o Len Scaling | 93.85 | 88.25 | 90.87 | 98.80 | 94.15 | 96.39 | 90.70 | 86.48 | 88.51 | 89.66 | 85.51 | 87.50 |
| Proposed | 93.84 | 88.48 | 90.98 | 98.78 | 94.18 | 96.39 | **90.79** | 86.62 | 88.63 | **89.78** | 85.68 | 87.65 |
| Proposed (batchsize=20) | 93.85 | 88.72 | **91.11** | 98.66 | 94.50 | 96.51 | 90.68 | **86.89** | **88.72** | 89.68 | **85.96** | **87.75** |
| MAESTRO v3 | | | | | | | | | | | | |
| Proposed (batchsize= 20) | 93.79 | 88.36 | 90.75 | 98.69 | 93.96 | 96.11 | 90.79 | 86.46 | 88.42 | 89.78 | 85.51 | 87.44 |

Table 1: Piano transcription note results for the proposed methods and various related works.

| Method | Activation | | | Onset | | | Onset & Offset | | |
|---|---|---|---|---|---|---|---|---|---|
| | P | R | $F_1$ | P | R | $F_1$ | P | R | $F_1$ |
| MAESTRO v2 | | | | | | | | | |
| Kong et al. [2020] | 94.14 | **94.29** | **94.11** | 77.43 | **78.19** | 77.71 | 73.56 | 74.21 | 73.81 |
| Proposed w/o Extra Win | 95.29 | 86.36 | 90.02 | 81.56 | 73.47 | 76.99 | 78.14 | 70.50 | 73.83 |
| Proposed w/o Post Conv | 95.19 | 87.05 | 90.36 | 81.72 | 73.29 | 76.99 | 77.99 | 70.05 | 73.54 |
| Proposed w/o Len Scaling | 94.73 | 87.32 | 90.34 | 80.79 | 73.51 | 76.71 | 77.36 | 70.51 | 73.53 |
| Proposed | 95.13 | 87.71 | 90.73 | 82.14 | 74.91 | 78.10 | 78.48 | 71.72 | 74.71 |
| Proposed (split) | 95.20 | 89.60 | 91.84 | **83.01** | 77.55 | **79.98** | **79.72** | **74.55** | **76.85** |
| Proposed (batchsize=20) | **95.35** | 87.63 | 90.78 | 82.27 | 75.61 | 78.55 | 78.72 | 72.42 | 75.20 |
| MAESTRO v3 | | | | | | | | | |
| Proposed (batchsize= 20) | 95.17 | 88.33 | 90.98 | 82.18 | 75.81 | 78.52 | 78.75 | 72.74 | 75.30 |

Table 2: Sustain pedal detection results for the proposed methods and various related works.

On MAESTRO v2, the proposed system achieves a *Note w/ Offset $F_1$* of 88.72% and a *Note w/ Offset & Velocity $F_1$* of 87.75%, significantly outperforming previous methods in predicting holistic note events. Regarding the *Note Onset* F1 score, Kong et al. [2020] has the highest performance. Our proposed method slightly underperforms Kong et al. [2020], but still outperforms Hawthorne et al. [2019] by about 1.5%. Regarding the *Activation* F1 score, our proposed method achieves the highest performance, outperforming Kong et al. [2020] by 1% and outperforming Hawthorne et al. [2019] by 3%. This is the case even though the proposed method is not optimized to output frame-level predictions, while the compared methods both have a frame-level prediction branch.

The *Proposed w/o Extra Wins* entry refers to the variation which does not apply extra adjustable windows for creating multiple version of log-mel spectrogram as the input. The results indicate that this oversampled log-mel spectrogram input helped improve the system slightly. The *Proposed w/o Post Conv* entry refers to the variation which does not include the final convolutional block when computing the score matrix. The results indicate that this convolutional block also helped improve the system slightly. The *Proposed w/o Len Scaling* entry refers to the variation which scales the scores in the intervallic score matrix by the length of the interval. The results indicate that this scaling of intervallic scores also helped improve the system slightly. We also found that increasing the batch size from 12 to 20 improves the result.

---

[2]Regarding discrepancies with respect to numbers reported in [Kong et al 2020], we inspected their code and found a bug that affects a small portion of note offsets when handling pedal extension. In our evaluation, we use the correct offset labels, which improves the results of their model slightly. For pedals, they use a onset tolerance of 200ms.

In terms of sustain pedal transcription results (Table 2), our method achieves an *Onset $F_1$* of 78.55% and an *Onset w/ Offset $F_1$* of 75.2%, outperforming Kong et al. [2020] in predicting holistic pedal events. Because Kong et al. [2020] uses a separate model replicating their note transcription model to predict pedal activity separately, we also experimented with splitting the original architecture into two branches with only the convolutional blocks shared, still training these two tasks (notes and pedals) together. The result is shown in the table as *Proposed (split)*. This variation with two branches for notes and pedals further improves all F1 metrics for pedal transcription. It suggests that there may be some negative transfer between these two tasks and further improvements may be achieved by tuning branches for notes and pedals separately. Kong et al. [2020] yields a $F_1$ of 94.11% for predicting the pedal activation, that is 2% higher than the proposed model, due to its higher recall. This difference may be due to the fact that our system is only trained to make predictions on the holistic events instead of individual frames.

## 4.5   Runtime Analysis

The proposed model has an $\mathcal{O}(L^2 N)$ time complexity, where $L$ is the length of the input sequence, i.e., the number of frames, and $N$ is the number of *eventType*(s). The semi-CRF layers involve sequential computation, and traditionally were often considered too slow for practical use with respect to long sequences if no restrictions were applied. Here we benchmark the computation time of several components of the proposed algorithm that have quadratic time complexity, using an input sequence of a reasonable length for music transcription. The results are shown in Table 3. With a hop size of 1024 samples and a sampling rate of 44.1 KHz, 400, 800, and 1600 frames correspond to audio segments of 9.29 s, 18.58 s, and 37.15 s, respectively, which are common lengths used for audio processing. For these lengths, the ratio of the audio length to the time taken by each component are within the acceptable range. The algorithms were implemented in *PyTorch*, and we believe that further speedup can be achieved with a native C++/CUDA implementation.

| Components | 400 Frames | | 800 Frames | | 1600 Frames | |
|---|---|---|---|---|---|---|
| | GPU | CPU | GPU | CPU | GPU | CPU |
| Forward Backward (*PyTorch* autodiff) | 0.39 | 5.95 | 4.40 | 44.69 | 17.68 | 354.40 |
| Forward Only | 0.05 | 0.20 | 0.09 | 0.80 | 0.23 | 3.08 |
| Forward-Backward | 0.08 | 0.58 | 0.12 | 2.32 | 0.27 | 7.12 |
| Viterbi | 0.23 | 0.16 | 0.27 | 0.36 | 0.54 | 0.88 |
| Pairwise Scores | 0.06 | 0.84 | 0.25 | 3.14 | 0.99 | 11.76 |

Table 3: Running time (seconds) of algorithm components that have quadratic time complexity w.r.t. the input length on Intel(R) Core(TM) i7-7800X CPU @ 3.50 GHz and Nvidia GTX 1080TI. Events for $N = 90$ *eventType*s and for a single audio segment are predicted.

To get a clearer idea on the overall speed of the system, we then compare the entire running time with the previous state-of-the-art system transcribing the same audio file, *Carl Czerny Grand Sonata Op.145 No.9*, which is 33.3 minutes long. For a fair comparison with Kong et al. [2020], we do not batchify our computations across audio segments for transcribing a single audio file. Our system takes 95s while Kong et al. [2020] takes 353s.

## 5   Conclusions

In this work, we proposed a piano transcription system designed to directly predict note events. It uses a specialized formulation of a semi-Markov conditional random fields, where the prediction targets are a set of non-overlapping events for each piano key and the sustain pedal. Using this formulation, we eliminate the need to build post-processing algorithms or heuristics atop frame-level estimates, as well as the need to aggregate disjoint predictions from multiple tasks, e.g., onset, offset and pitch estimation. The results of our experiments on MAESTRO, a popular piano transcription dataset, show that our model significantly outperforms two state-of-the-art piano transcription methods on note-level transcription. We show that the proposed method can be implemented efficiently enough for the task domain and that the final system transcribes faster than the existing state of the art system. We believe that this simple, fast, and well-performing approach is extensible to other similar tasks which currently rely on frame-level estimates.

# 6 Funding Disclosure

This work has been partially funded by the National Science Foundation grants No. 184614 and DGE-1922591. Additional revenues related to this work: Zhiyao Duan's sabbatical leave at Kwai Inc. and part-time activities at Mango Future Group Limited.

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
