

Figure 3: Adjustible Gaussian windows after training.

## A    Note on adjustible windows for producing oversampled spectrogram

As mentioned in the main text, we experimented with applying multiple window functions to create a log-mel spectrogram input, with multiple channels (for each window function) on each time-frequency bin. In this work, the two-parameter adjustible Gaussian window is parameterized as follows:

$$w[i] = \exp\left[-0.5\left(\frac{i - N\sigma(p_1)}{N\sigma(p_2)/2}\right)^2\right], \quad i = 0, 1, \ldots, N - 1 \tag{6}$$

where $N$ is the size of the window, $p_1$ and $p_2$ are two parameters for the center and the standard deviation, respectively, and $\sigma(\cdot)$ is the sigmoid function which maps the parameters to be between (0,1). Here, $p_1$ is initialized such that centers are equidistantly placed between 0 and 1, and $p_2$ is initialized to $-1$. The final trained window bank is shown in Fig. 3.

## B    Note on activation-level evaluation metrics

For the commonly reported frame-level metrics (precision, recall, $F_1$), we identify that the actual implementations vary from paper to paper, with different hop sizes, e.g., Hawthorne et al. [2018, 2019], Kong et al. [2020]. In Hawthorne et al. [2019, 2018], the authors use a hop size that is nearly twice the one used in Kong et al. [2020]. Also these metrics, in the literature, are sometimes computed on the raw grid of frame-level predictions, which does not reflect the final transcription quality. These two facts render these metrics ill-suited for comparing systems. In order to make the evaluation metrics more general, emphasizing the temporal overlap between the predicted and ground truth time spans for activations of specific events, we propose to directly evaluate event activations in continuous time on the final output of all MIDI transcription systems, i.e., the transcribed note events in continuous time, without the need for quantization. We denote these metrics as *activation-level metrics*, and define them as follows:

$$precision = \frac{total\ duration\ correctly\ predicted}{total\ duration\ predicted}$$

$$recall = \frac{total\ duration\ correctly\ predicted}{total\ duration\ in\ ground\ truth} \tag{7}$$

$$F_1 = 2\left(\frac{precision * recall}{precision + recall}\right)$$

Here the total duration correctly predicted is the summation of all the durations of the overlapping regions between the predicted time spans and the ground truth time spans.

# C    Timing Precision

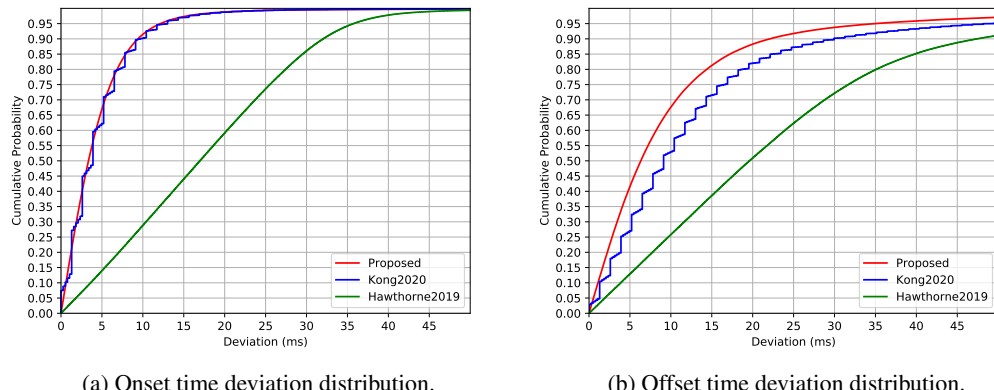

(a) Onset time deviation distribution.     (b) Offset time deviation distribution.

Figure 4: Empirical cumulative distribution functions of time deviations of estimated onsets and offsets from ground-truth notes.

We compare the proposed system to two state-of-the-art methods w.r.t. timing precision of onset and offset estimation among correctly transcribed notes. Here the matching criteria is relaxed to a tolerance of 100 ms for onsets and the greater value of 100 ms and 20% of the note duration for offsets. The results are shown in Fig. 4 in the form of empirical cumulative distribution functions (ECDFs). The frame hop sizes used by the three methods shown in the figure are 23ms, 10ms, 32 ms, respectively. Regarding onsets, the proposed method and Kong et al. [2020] perform the best, showing 90% of deviations are less than 10 ms, while Hawthorne et al. [2019] shows a uniform distribution below 32 ms, which is the frame hop size it uses. The curve of Kong et al. [2020] is not smooth, due to the peak interpolation strategy it adopts. Regarding offsets, the proposed method outperforms both comparison methods. In particular, 90% of offset deviations are less than 23 ms, while the value is 30 ms for Kong et al. [2020] and 47 ms for Hawthorne et al. [2019]. Our strategy of regressing to dequantize positions reaches a similar or better performance compared with Kong et al. [2020], even with more than twice the hop size. Our method also produces a smooth ECDF curve.