# OpenReview forum: "Skipping the Frame-Level: Event-Based Piano Transcription With Neural Semi-CRFs"
_NeurIPS.cc/2021/Conference — NeurIPS 2021 Poster_

### Official Review · Reviewer_LiUp · 2021-07-13

**Rating:** 4
**Confidence:** 3

**Summary:**

They propose an "event-based prediction" (instead of relying on frame-level estimates) for piano transcription. They formalize this idea via a semi-markov conditional random fields (Semi-CRF) – since, in the author's words, "the Semi-CRF layer promotes a direct formulation for event (interval) prediction, eliminating the need for estimating activations at the frame-level followed by post processing".

**Limitations And Societal Impact:**

Limitations: the model it's not very novel from the machine learning perspective, and the results do not clearly outperform the state-of-the-art.

Societal Impact: music transcription, if solved, could be a very useful intermediate tool for music information retrieval. Together with music source separation, are considered the "holy grail" since solving these two tasks could help many relevant tasks and applications.

**Main Review:**

Their method obtains comparable results to the current state-of-the-art, arguing that their approach is faster than state-of-the-art. However they do not report run times for the state-of-the-art models for a fair comparison.

I also don't couldn't identify substantial machine learning contribution, because the method they employ (semi-CRF) were already proposed in 2004. Further, their empirical results are not significantly improving current state-of-the-art performance. However, the idea of using semi-CRF for this tasks (relying on events instead of frame-level estimates) is novel and interesting. For this reason, I'd recommend to submit into a music (e.g., ISMIR) or audio (e.g., ICASSP) research venues.

Minor comments:
- Line 16: "while still quadratic in complexity" -> this implies a (not presented nor mentioned) baseline that is quadratic in complexity.
- Line 114: "(See Section XXX)".
- It was unclear when you were referring to "non-overlapping". Overlap in time or in frequency or pitches overlap?

**Time Spent Reviewing:**

2

---

> ### Author Response · Authors · 2021-08-10
> **Thanks for your review  and suggestions.**
>
> Thanks for your review  and suggestions.
> We added the running comparison in our common response, B.2. Please take a look.
> The significant improvement over the state-of-the-art model is on the holistic note predictions.
> With a similar model architecture (our contextual model is largely the same as one single head used in [Kong et al 2020]), it is highly likely that optimal results on frame-level metrics ((frame) activations, onsets)will be achieved by a model optimized to do so..
>
> In our updated results, B.1. in the common response, we show that the proposed system outperforms on all event-level F1s  and on the frame (activation)-level f1 for note transcription.
>
> Indeed, the correct (within tolerance) pitch/onset/offset metric and pitch/onset/offset/velocity metric are precisely what we aim to improve. The other metrics, such as pitch/onset, while commonly used to evaluate music transcription systems, do not consider notes holistically. In the past, the note-level metrics with offset have been relatively overlooked, but we want to stress that without offset, the duration of a note is not taken into account during evaluation. For the most important applications of automatic music transcription, duration, not only onset and pitch, is crucial for the quality of transcribed pieces in music notation and sound reproduction.
>
>
> Also, we provide our response on novelty in our common response. Please take a look.

---

### Official Review · Reviewer_ygBT · 2021-07-15

**Rating:** 5
**Confidence:** 5

**Summary:**

The paper proposes to use semi-CRF for piano transcription. Since piano transcripts include asynchronous events and vanilla semi-CRF is not able to cope with these, the paper assumes that the events are independent from each other. Additional features for predicting the velocity, onset, offset refinements are included. Experiments show that the semi-CRF approach can be on par with the state of the art.

**Limitations And Societal Impact:**

Perhaps the system can be used to judge ones piano performance, and that could have a negative societal impact.

**Main Review:**

I am giving a score of 5, because the novelty is somewhat lacking, the approach assumes a strong assumption, the experiments do not demonstrate the strengths, weaknesses, properties of the proposed model.

# Novelty

The novelty of this paper lies in the application of semi-CRF to the task of piano transcription, so it is somewhat lacking. Piano transcription is actually the perfect task to extend semi-CRF to handle asynchronous events. The novelty could have been a lot more significant, if the paper extended semi-CRF for that.

# Approach

The major assumption in the approach is that the events are assumed to be independent of each other. This is clearly not the right assumption, but it is somewhat alleviated by the fact that the scores are generated by a one network. In other words, it can be seen as 88 + 3 independent semi-CRF models with a shared feature encoder.

The paper also talks about the implementation in depth. It reads like an ad-hoc implementation, but in fact, the forward-backward can be implemented as matrix multiplications. A general implementation, such as torch-struct (link below), should suffice.

https://github.com/harvardnlp/pytorch-struct

Another approach is to implement forward-backward on CPU, while keeping the rest of the computation for neural networks on GPU. Comparing to the FLOPS in neural networks, the forward-backward in semi-CRF should not be the bottleneck.

Maximum duration is not imposed in this word, but I suppose if efficiency is of importance, maximum duration should be imposed. It might also help modeling, because it is much less likely to hold a pitch or a pedal for a really long time. It would be even better to have explicit duration modeling, and that the one of the strengths of using semi-CRF instead of other frame-based models.

# Experiments

The numbers for different systems are extremely close. I wonder what amount of improvement is considered significant. It seems that (Kong et al., 2020) is the better ones in most category. The paper does not provide any experiments to study the internals of the proposed model. If the goal is to get the best numbers, taking features from (Kong et al., 2020) and integrate them into semi-CRF might be worth a try.

The computation time table does not have a baseline. It is probably better to report real-time factors. Since one of the strengths of choosing the particular score function is its efficiency over (Kong et al., 2020), it is probably better to put the runtime of both models in the table.

# Presentation

The paper overall reads well.

Figure 1 is certainly helpful. However, it is probably better to write out the explicit forward computation with equations.

Algorithm 1 and 2 are presented in many other papers, for example (Kong et al., 2015). It might be better to save the space for more experiments or more analysis.

I'm a little surprised that the application of semi-CRF on speech is not mentioned in the related work. Speech input is actually more similar to the input for piano transcription, than to say named-entity recognition. See for example the paper below and the citations therein.

End-to-end neural segmental models for speech recognition
Hao Tang, Liang Lu, Lingpeng Kong, Kevin Gimpel, Karen Livescu, Chris Dyer, Noah A. Smith, Steve Renals
IEEE Journal of Selected Topics in Signal Processing, 2017


**Time Spent Reviewing:**

6

---

> ### Author Response · Authors · 2021-08-10
> **Thanks for your review and suggestions.**
>
> Thanks for your review and suggestions. We have given our response on the novelty of this paper in our common response; please take a look.
>
> We provide some justification on the independence assumption in our response to reviewer 1vpu. Please take a look.
>
> > The paper also talks about the implementation in depth. It reads like an ad-hoc implementation, but in fact, the forward-backward can be implemented as matrix multiplications. A general implementation, such as torch-struct (link below), should suffice.
> >https://github.com/harvardnlp/pytorch-struct
>
>
> We have already implemented the forward-backward using matrix multiplications.
> We also benchmarked the general semiCRF implementation in pytorch-struct as you provided.  For a typical size (use the same variable name in its documentation, batch=88, N=400, C=2, K = 400, which are batch size, number of steps, classes, maximum duration, respectively) for a single 10s audio segment, it directly triggers an OOM (out of memory) exception on both CPU and GPU for the log-partition function. We managed to make it work with a maximum duration of 100 on CPU (6.8s for the log-partition function) and additionally split along the batch dimension on GPU(0.28s for the log-partition function).  It’s significantly slower than our implementation (0.2s and 0.05s respectively without a duration limit).
>
>
> >Another approach is to implement forward-backward on CPU, while keeping the rest of the computation for neural networks on GPU. Compared to the FLOPS in neural networks, the forward-backward in semi-CRF should not be the bottleneck.
>
>
> In our benchmark, we showed that in a typical size, GPU works faster. The time taken by the semiCRF layer is not that trivial for one with average hardware resources.
>
> >Maximum duration is not imposed in this word, but I suppose if efficiency is of importance, maximum duration should be imposed. It might also help modeling, because it is much less likely to hold a pitch or a pedal for a really long time. It would be even better to have explicit duration modeling, and that the one of the strengths of using semi-CRF instead of other frame-based models.
>
> For the problem domain, durations for each note event have a large variety.  It’s not desirable for us to set a small threshold for cutting off certain events. If it is an issue, different pitches and pedals can have different maximum duration. On the other hand, a small limit on duration will limit the applicability of the proposed approach to other similar tasks.Thankfully, this is not an issue for our system.
>
> > The numbers for different systems are extremely close. I wonder what amount of improvement is considered significant. It seems that (Kong et al., 2020) is the better ones in most category. The paper does not provide any experiments to study the internals of the proposed model. If the goal is to get the best numbers, taking features from (Kong et al., 2020) and integrate them into semi-CRF might be worth a try.
>
> The model architecture is already intentionally designed to be largely similar to a single head of  [Kong et al., 2020] because the purpose of this paper is not to get all the best numbers, otherwise we can easily make some tuning on the architecture and apply tricks.
> Firstly, one should note that all baseline systems are optimized directly for predicting onset/frames which gives them advantage over metrics on these two things(frame(activation) / onset).
> Also note that [Kong et al., 2020] has tuned a parameter that has a similar effect as label smoothing and also they have a detecting threshold as a hyperparameter, which contributes to the slightly higher note onset F1 compared to our system. We can also tune these hyperparameters for our system but certain tuning is not desirable for our paper.
> In our updated results, B.1. of the common response, we show that the proposed system now outperforms on all event-level F1s  and on the frame (activation)-level f1 for note transcription.
> The difference in pedal frame(activation) F1 demonstrates the discrepancies on two different approaches: one is optimizing for frame-wise predictions and another is for predicting the holistic event.
>
> Indeed, the correct (within tolerance) pitch/onset/offset metric and pitch/onset/offset/velocity metric are precisely what we aim to improve. These two metrics evaluate the final output of the system holistically.
> The other metrics, such as pitch/onset, while commonly used to evaluate music transcription systems, do not consider notes holistically. In the past, the note-level metrics with offset have been relatively overlooked, but we want to stress that without offset, the duration of a note is not taken into account during evaluation. For the most important applications of automatic music transcription, duration, not only onset and pitch, is crucial for the quality of transcribed pieces in music notation and sound reproduction.
>
> > The computation time table does not have a baseline. It is probably better to report real-time factors. Since one of the strengths of choosing the particular score function is its efficiency over (Kong et al., 2020), it is probably better to put the runtime of both models in the table.
>
> We have added this comparison, see the B.2 in the common response.
>
>
> > Algorithm 1 and 2 are presented in many other papers, for example (Kong et al., 2015). It might be better to save the space for more experiments or more analysis.
>
> We include it here for one reason: our formulation is not the standard semiCRF: it’s specifically tailored for representing polyphonic events in continuous time. We include these pseudo-codes  in the paper because we do not assume all readers have the expertise to derive it from the specification.
>
> > I'm a little surprised that the application of semi-CRF on speech is not mentioned in the related work. Speech input is actually more similar to the input for piano transcription, than to say named-entity recognition. See for example the paper below and the citations therein.
> End-to-end neural segmental models for speech recognition Hao Tang, Liang Lu, Lingpeng Kong, Kevin Gimpel, Karen Livescu, Chris Dyer, Noah A. Smith, Steve Renals IEEE Journal of Selected Topics in Signal Processing, 2017
>
> We will cite this paper as well. In fact we have already cited one paper on the same task [kong 2015].

---

### Official Review · Reviewer_nVTb · 2021-07-16

**Rating:** 6
**Confidence:** 3

**Summary:**

This paper discusses a novel approach to piano transcription which involves predicting notes as events rather than frame-level activity. This approach aligns well with the eventual goal of music transcription, i.e., transcribing musical audio to some form of music notation, e.g.: MIDI or staff which are both based on note events. The method proposed utilizes semi-Markov CRFs which have been utilized to directly predict note/pedal events as non-overlapping continuous intervals in the audio. Utilizing standard metrics for piano transcription, the authors show that their method outperforms current state-of-the-art methods which perform frame-wise transcription.

**Limitations And Societal Impact:**

The authors do not discuss any potential negative societal impacts of their work. I am unable to think of negative impacts of algorithms for piano transcription. Perhaps the authors can discuss the impact of automatic music transcription (AMT) algorithms broadly. For example, musicians oftentimes teach lessons or sell transcriptions of their works (e.g. via Sheet Happens Publishing) and AMT systems can potentially cut into their earnings if used maliciously.

**Main Review:**

Strengths:
- The approach presented in this paper steps away from recent method for piano transcription which are often improvements of the Onsets and Frames paper by Hawthorne et al. This approach paves a way for modern event-based transcription and may inspire more researchers to approach similar problems using the framework shown here.
- The results of this approach are better than current SOTA which are for frame-based transcription.
- The method is well explained and evaluated using the standard dataset (MAESTRO) and metrics.

Weaknesses:
- I do not see many issues with this paper in terms of methodology. One downside to this paper can possibly be w.r.t broader interest in the ML community. The authors do a good job of framing an important problem in the music information retrieval space in the framework of semi-Markov CRFs which is novel, but in terms of algorithmic contribution, I am not sure if there is much novelty. This paper might better suit the audience of conferences such as ICASSP or ISMIR since the scope is pretty narrow.

Overall Impression:
- As mentioned above, this paper presents a novel approach to solving the piano transcription problem. The method is well explained and evaluated thoroughly leading to improvements over current SOTA. My only criticism towards this paper in the context of this conference is that perhaps the contribution is too narrow in scope. Therefore I rate this paper marginally above acceptance.

Comments:
- Line 114: Section XXX -> Section 3.3
- The writing can be organized a little better mainly in Section 3. There is some mixup of algorithm details and implementation details which might be better left separate. This is simply a stylistic suggestion. Feel free to ignore.


**Time Spent Reviewing:**

6

---

> ### Author Response · Authors · 2021-08-10
> **Thanks for your review and suggestions**
>
> Thanks for your review and suggestions. We provide a response on novelty in the common response, A.1. Please take a look.

---

### Official Review · Reviewer_1vpu · 2021-07-16

**Rating:** 5
**Confidence:** 4

**Summary:**

This paper applies a semi-CRF to the music transcription task, with experimental results on the MAESTRO piano dataset.

**Main Review:**

I found this paper difficult to read, to an extent that challenged my ability understand all the technical details. I will do my best to review the content below, with an understanding that I may have misunderstood certain aspects of this paper. I have a strong background in music transcription, and some familiarity with the mathematics of CRF's, so I believe this paper is objectively quite confusing. For that reason alone, I recommend against its publication in this current form.

To the best of my understanding, this paper does not contain a significant contribution to the Semi-CRF literature. Therefore, it should be evaluated on the strength of its contribution to music transcription. The contribution to transcription has two components: (1) a reframing of the transcription task that "skips the frame-level" and (2) reasonably strong empirical transcription results on the MAESTRO dataset. I discuss the empirical results at the end of the review.



### The Semi-CRF Implementation

I like the framing of music transcription as a sparse interval labeling problem. It less clear to me why this Semi-CRF implementation is a useful way to accomplish this. The Semi-CRF described in this paper is crippled in two significant ways that seem to undermine the advantages of using such a model. First, an independent Semi-CRF is constructed for each event type (pitch class) so by design this model does not model note correlations. Second, this implementation is a zeroth order Semi-CRF so it also does not model temporal correlations. I understand that these design restrictions may be necessary to adapt the problem to a Semi-CRF's structure, but I would have liked to see some motivating discussions for the decision to use a Semi-CRF, versus e.g. the seq2seq models that have replaced CRF's in many problem domains.


I don't understand what the parameters "theta" are in Equation 1. A traditional CRF is a log-linear model, so I initially guessed that "theta" was a set of weights on pre-trained feature vectors generated by the score and score_eps models. But on closer reading, I think these are actually the parameters of the score models themselves? In that case, I guess you are training the CRF end-to-end with the neural score and contextual models? That suggests to me that you are computing gradients via some hybrid approach, wherein you compute partial derivatives w.r.t. the neural model outputs via analytical expressions (Algorithm 1) and then chain these calculations together with derivatives of the neural model outputs w.r.t. their parameters. Whether or not this guess is correct, the current description is very confusing!

Related to these questions about the parameters and calculation of gradients: I would also be interested if the authors could provide any insight about the claim (lines 137-139) that Algorithm 1 is faster than autodiff. Autodiff should be a small constant factor times the cost of a forward pass, so unless there really is just a big win on small constants here (e.g. 0-2x analytical versus 4-5x autodiff) this claim is somewhat surprising. Is this a well-established fact in the CRF literature that analytical gradients are faster than autodiff?


I don't entirely understand the asymmetry in handling event (score) versus no-event (score_eps) labels. Rather than having these "empty" no-event positions, couldn't we use a dense labeling of event and no-event intervals? This seems like it would be more efficient too: rather than evaluating score_eps at every unlabeled frame, we would make just one calculation for each contiguous chunk of time without an event. If this isn't possible, then there should be a clearer discussion of why this modeling choice was necessary.


It looks like there is a single contextual model shared by all the upstream score and attribute models. And there are distinct score and attribute models for each event type that do not share weights with each other. Is that correct?


### Empirical Results


I find the empirical state-of-the-art claims slightly overstated and underanalyzed. Results are significantly better than the Kong et al. baseline for Note w/ Offset and Note w/ Offset & Velocity metrics, but only marginally better for the frame wise metric (90.81 vs 90.17) and marginally worse under note onset measurement (96.31 vs 96.77). Sustain pedal detection is significantly worse than the Kong et al. baseline. It makes some intuitive sense to me that the note-based metrics might favor the Semi-CRF, but the paper doesn't make an argument about why we should objectively prefer these metrics. I would have liked to see an analysis about the source of disagreement among these metrics (which a priori, we might expect to be very correlated). This might provide evidence for why we should prefer a certain type of metric, and potentially support the SOTA claims that only really hold up for the note-based metrics.


### Post-Rebuttal and Discussion


Thanks for your detailed response. I appreciate the quantitative analysis of claims about analytical vs. autodiff implementations. There is some lingering confusion among the reviewers (myself included) about *why* this would be the case, and why the DP implementation is the bottleneck in these systems. Further analysis of these questions (in addition to the empirical statistics) could bolster the machine learning contribution of this work and complement the music contribution. This might help with selling this work to the ML community.

Your response about temporal and harmonic correlations is reasonable, and I think it would really help to add this as motivating discussion to the paper (possibly even in the introduction). This would help to clarify what you're doing, to justify why you're doing it this way, and to avoid surprising reviewers/readers with independence assumptions later on.

I have increased my score to a 5, in light of (1) the stronger empirical transcription results, (2) the empirical support for claims about the DP implementation, and (3) promises of improvements to the paper's clarity & exposition. Because I can't evaluate the extent of the writing improvements (3), I am reluctant to give a stronger endorsement to the current version. But I hope you don't take my review too negatively: I think this is good work, and with some more polish I think this will become a very good paper.

**Time Spent Reviewing:**

5

---

> ### Author Response · Authors · 2021-08-10
> **Thanks for your review and suggestions.**
>
> Thanks for your review and suggestions. We have greatly improved the clarity/readability of the paper since the initial submission (however, it’s not allowed to upload the revision at this time).
>
> > First, an independent Semi-CRF is constructed for each event type (pitch class) so by design this model does not model note correlations. Second, this implementation is a zeroth order Semi-CRF so it also does not model temporal correlations
>
> An expressive model for the correlations in the data space (temporal correlations and correlation between simultaneous notes) faces the problem of imposing biases towards specific styles and conditions a piano is being played.  We view piano transcription as an inverse problem for estimating the input to the piano from the sound produced by the piano. Zeroth-order semiCRF models the minimal temporal correlation: it imposes the constraint that each key (pitch) is a monophonic source. It’s true that the independent semiCRFs do not encode the correlations between notes being played simultaneously, neither do most SOTA works on piano transcription. It’s currently unclear whether or not modeling these correlations help: we also conjecture it would also be an evaluation metric-dependent thing. We leave this for future exploration.
>
> >I don't understand what the parameters "theta" are in Equation 1. A traditional CRF is a log-linear model, so I initially guessed that "theta" was a set of weights on pre-trained feature vectors generated by the score and score_eps models. But on closer reading, I think these are actually the parameters of the score models themselves? In that case, I guess you are training the CRF end-to-end with the neural score and contextual models? That suggests to me that you are computing gradients via some hybrid approach, wherein you compute partial derivatives w.r.t. the neural model outputs via analytical expressions (Algorithm 1) and then chain these calculations together with derivatives of the neural model outputs w.r.t. their parameters. Whether or not this guess is correct, the current description is very confusing!
>
> It’s correct that the entire model is trained end-to-end. We will improve the clarity of our paper on this.
>
> > Related to these questions about the parameters and calculation of gradients: I would also be interested if the authors could provide any insight about the claim (lines 137-139) that Algorithm 1 is faster than autodiff. Autodiff should be a small constant factor times the cost of a forward pass, so unless there really is just a big win on small constants here (e.g. 0-2x analytical versus 4-5x autodiff) this claim is somewhat surprising. Is this a well-established fact in the CRF literature that analytical gradients are faster than autodiff?
>
> It’s true that, in theory, autodiff should be a small constant times the forward pass. However, in reality, some other factors like memory management may dominate for a task domain with large size. We added a benchmark for the autodiff version of the forward-backward in the paper:
>
> |  | 400 |  | 800 |  | 1600 |  |
> |---|---|---|---|---|---|---|
> |  | GPU | CPU | GPU | CPU | GPU | CPU |
> | Forward | 0.05 | 0.20 | 0.09 | 0.8 | 0.23 | 3.08 |
> | Forward-backward | 0.08 | 0.58 | 0.12 | 2.32 | 0.27 | 7.12 |
> | Pytorch Autodiff | 0.39 | 5.95 | 4.40 | 44.69 | 17.68 | 354.40 |
>
> From the table we can see the analytical only requires an additional 0.6x vs autodiff 6.8x for 400 frames (GPU), and the difference is getting larger with larger length.
>
> > I don't entirely understand the asymmetry in handling event (score) versus no-event (score_eps) labels. Rather than having these "empty" no-event positions, couldn't we use a dense labeling of event and no-event intervals? This seems like it would be more efficient too: rather than evaluating score_eps at every unlabeled frame, we would make just one calculation for each contiguous chunk of time without an event. If this isn't possible, then there should be a clearer discussion of why this modeling choice was necessary.
>
> For the score_eps, we will add more explanation in the paper.  For no-event intervals, if the formulation is “symmetric” in your word, then the no-event intervals will have many non-unique representations. That’s what we want to avoid. Another consideration for score_eps is that it allows the model degenerate to make framewise predictions.
>
> > It looks like there is a single contextual model shared by all the upstream score and attribute models. And there are distinct score and attribute models for each event type that do not share weights with each other. Is that correct?
>
> Score and attribute models are also shared: the score model is essentially a NN with multiple channel output; the attribute model takes an eventType token as input.
>
> > I find the empirical state-of-the-art claims slightly overstated and underanalyzed. Results are significantly better than the Kong et al. baseline for Note w/ Offset and Note w/ Offset & Velocity metrics, but only marginally better for the frame wise metric (90.81 vs 90.17) and marginally worse under note onset measurement (96.31 vs 96.77). Sustain pedal detection is significantly worse than the Kong et al. baseline. It makes some intuitive sense to me that the note-based metrics might favor the Semi-CRF, but the paper doesn't make an argument about why we should objectively prefer these metrics. I would have liked to see an analysis about the source of disagreement among these metrics (which a priori, we might expect to be very correlated). This might provide evidence for why we should prefer a certain type of metric, and potentially support the SOTA claims that only really hold up for the note-based metrics.
>
> We updated our numbers (B.1. in the common response) after fixing one bug. The sustain pedal now also outperforms on the event-level F1s.  We have noticed the significant difference for the pedal frame (activation)-level metrics. This difference demonstrates the discrepancies between the two approaches. Indeed, the correct (within tolerance) pitch/onset/offset metric and pitch/onset/offset/velocity metric are precisely what we aim to improve. The other metrics, such as pitch/onset, while commonly used to evaluate music transcription systems, do not consider notes holistically. In the past, the note-level metrics with offset have been relatively overlooked, but we want to stress that without offset, the duration of a note is not taken into account during evaluation. For the most important applications of automatic music transcription, duration, not only onset and pitch, is crucial for the quality of transcribed pieces in music notation and sound reproduction.

---

### Author Response · Authors · 2021-08-10
**Common response**

Thanks for all the reviewers' efforts in reviewing the paper. We’ve created this individual post to clarify common concerns.

## A


1. Novelty/Originality

We did anticipate some uncertainty or dispute over the novelty of this work. On the surface, this work is a fusion of existing neural network architectures with a semiCRF layer. It’s true that semiCRFs have long been applied to a lot of areas. Even the famous line breaking algorithm [Knuth&Plass 1981] in Tex, the earliest example to the best of our knowledge, is actually performing semiCRF decoding.

The novelty of our work can be argued from several perspectives. On one hand, for the application domain, as the complexity of models grows larger and larger, our method is on the simple side. It is a straightforward formulation of event estimation and an overall improvement and simplification of the entire piano transcription pipeline. Aside from the intrinsic value of simplicity itself, our work is the first attempt at viewing each single event (note) for the application domain as a whole, and we demonstrate that our system is both better and faster compared to current state of the art systems. Our method for viewing events holistically may also be applicable to other similar tasks like polyphonic audio events detection, since this task is also typically formulated as the aggregation of frame-level predictions. Regarding the novelty of the proposed semiCRF formulation, aside from the proposed adaptation of the semiCRF for modeling simultaneous events that happen in continuous time, the size of the semi-CRF layer for this specific application domain is, by far, larger than any published work. For example, transcribing an audio segment with 400 frames (~10s) involves the simultaneous decoding of at least 88 SemiCRFs with length 400 and with no explicit upper limit on the event duration. With this in mind, we demonstrate that a semiCRF layer at this scale (in the paper we also benchmarked the layer up to length 1600) can actually be quite efficient through our implementation, contrary to traditional thinking, and thus can be popularized through this work for similar tasks.



## B

Here we list some improvements and revisions made since the initial submission:

1. We fixed one bug that only influenced the first audio segment in an audio file during training for our model. Results are boosted slightly. The proposed method now outperforms the baselines for all note-level metrics (P, R, F1).
    |  | Activation |  |  | Note Onset |  |  | Note w/Offset |  |  | Note w/Offset & Vel. |  |  |
|---|---|---|---|---|---|---|---|---|---|---|---|---|
| Method | P | R | F1 | P | R | F1 | P | R | F1 | P | R | F1 |
| Hawthorne et al. [2019] | 86.84 | 89.24 | 87.82 | 97.88 | 92.26 | 94.93 | 82.09 | 77.44 | 79.65 | 78.37 | 73.94 | 76.05 |
| Kong et al. [2020] | 90.09 | 90.42 | 90.15 | 98.16 | 95.46 | 96.77 | 85.65 | 83.32 | 84.45 | 84.18 | 81.92 | 83.02 |
| Proposed | 93.84 | 88.48 | 90.98 | 98.78 | 94.18 | 96.39 | 90.79 | 86.62 | 88.63 | 89.78 | 85.68 | 87.65 |

And for pedals:

|  | Activation |  |  | Onset |  |  | Onset w/Offset |  |  |
|---|---|---|---|---|---|---|---|---|---|
| Method | P | R | F1 | P | R | F1 | P | R | F1 |
| Kong et al. [2020] | 94.14 | 94.29 | 94.11 | 77.43 | 78.19 | 77.71 | 73.56 | 74.21 | 73.81 |
| Proposed | 95.13 | 87.71 | 90.73 | 82.14 | 74.91 | 78.10 | 78.48 | 71.72 | 74.71 |


2. We added a running time comparison for transcribing the same 33.3 minute-long piano performance: for [Kong 2020] it takes 353s while the proposed system only takes 95s. Note that for a fair comparison, we do not batch the inference over audio segments since this is not done in [Kong 2020].

3. We have improved the overall readability of the paper.

---

### Decision · Program_Chairs · 2021-09-28

**Decision:**

Accept (Poster)

**Comment:**

This paper investigates piano transcription based on semi-CRF models.  This approach views music note labeling as an event prediction problem.  The semi-CRF is used to score the non-overlapping continuous intervals to avoid conventionally used frame-level local decision.   Experiments are conducted on the MAESTRO  dataset and competing results are reported by the authors.   All reviewers agree that the idea of applying semi-CRF to music transcription is novel to the application domain but raise concerns on the implementation of semi-CRF.  For instance, temporal correlation of events is not considered in the current semi-CRF model.  Also, it is not clear why autodiff loses advantages over the authors' implementation in speed.  The rebuttal is good which answers the majority of the questions.  Overall, the idea seems to be interesting to the music transcription but the revision to be made in responses to the concerns seems to be significant. Given its current form and some lingering concerns, I would recommend the authors revise and resubmit the work to a future conference.

**Consistency Experiment:**

NeurIPS has a long history of experimentation. In 2014, NeurIPS ran an experiment in which 10% of submissions were reviewed by two independent committees to quantify the randomness in the review process. This year, we repeated a variant of this experiment to see how the quality of the review process has changed over time.  This paper was part of the experiment and was therefore assigned to two committees (consisting of reviewers, an Area Chair, and a Senior Area Chair) that reached independent decisions.  If both committees made the same recommendation, this recommendation was followed. If a single committee recommended acceptance, the paper was accepted (with the exception of a few cases in which the other committee identified what we considered a fatal flaw, e.g., an error in a key result).

This copy’s committee reached the following decision: **Reject**

The other committee assigned to the paper recommended **Accept (Poster)**.  You can find the other set of reviews, along with any follow up discussion with the authors here:
https://openreview.net/forum?id=DGA8XbJ8FVd